# Tumor Growth Suppression of Pancreatic Cancer Orthotopic Xenograft Model by CEA-Targeting CAR-T Cells

**DOI:** 10.3390/cancers15030601

**Published:** 2023-01-18

**Authors:** Osamu Sato, Takahiro Tsuchikawa, Takuma Kato, Yasunori Amaishi, Sachiko Okamoto, Junichi Mineno, Yuta Takeuchi, Katsunori Sasaki, Toru Nakamura, Kazufumi Umemoto, Tomohiro Suzuki, Linan Wang, Yizheng Wang, Kanako C. Hatanaka, Tomoko Mitsuhashi, Yutaka Hatanaka, Hiroshi Shiku, Satoshi Hirano

**Affiliations:** 1Department of Gastroenterological Surgery II, Hokkaido University Faculty of Medicine, Sapporo 060-8638, Hokkaido, Japan; 2Department of Cellular and Molecular Immunology, Mie University Graduate School of Medicine, Tsu 514-8507, Mie, Japan; 3Center for Comprehensive Cancer Immunotherapy, Mie University, Tsu 514-8507, Mie, Japan; 4Takara Bio Inc., Kusatsu 525-0058, Shiga, Japan; 5Department of Immuno-Gene Therapy, Mie University Graduate School of Medicine, Tsu 514-8507, Mie, Japan; 6Department of Personalized Cancer Immunotherapy, Mie University Graduate School of Medicine, Tsu 514-8507, Mie, Japan; 7Research Division of Genome Companion Diagnostics, Hokkaido University Hospital, Sapporo 060-8648, Hokkaido, Japan; 8Department of Surgical Pathology, Hokkaido University Hospital, Sapporo 060-8648, Hokkaido, Japan

**Keywords:** chimeric antigen receptor engineered T cell, carcinoembryonic antigen, pancreatic ductal carcinoma, adoptive cell therapy, orthotopic xenograft mouse model

## Abstract

**Simple Summary:**

Pancreatic ductal adenocarcinoma is one of the most lethal malignancies, and there are vast unmet medical needs. In this study, we hypothesized that chimeric antigen receptor engineered T cell (CAR-T) targeting carcinoembryonic antigen (CEA) would be effective in the treatment of pancreatic ductal adenocarcinoma. In vivo experiments in a more clinically similar environment were considered necessary; we examined the antitumor effects of adoptive anti-CEA-CAR-T, using orthotopic xenograft mouse models of pancreatic ductal adenocarcinoma. As result, the therapeutic effect of anti-CEA-CAR-T therapy was related to the CEA expression level. Furthermore, the retrospective analysis of pathological findings from pancreatic ductal adenocarcinoma patients showed a correlation between the intensity of CEA immunostaining and tumor heterogeneity. These findings show that anti-CEA-CAR-T therapy can be useful for pancreatic ductal adenocarcinoma; furthermore, the pathological findings of CEA can be clinically used as biomarkers to select cases for anti-CEA-CAR-T therapy.

**Abstract:**

Chimeric antigen receptor engineered T cell (CAR-T) therapy has high therapeutic efficacy against blood cancers, but it has not shown satisfactory results in solid tumors. Therefore, we examined the therapeutic effect of CAR-T therapy targeting carcinoembryonic antigen (CEA) in pancreatic adenocarcinoma (PDAC). CEA expression levels on the cell membranes of various PDAC cell lines were evaluated using flow cytometry and the cells were divided into high, medium, and low expression groups. The relationship between CEA expression level and the antitumor effect of anti-CEA-CAR-T was evaluated using a functional assay for various PDAC cell lines; a significant correlation was observed between CEA expression level and the antitumor effect. We created orthotopic PDAC xenograft mouse models and injected with anti-CEA-CAR-T; only the cell line with high CEA expression exhibited a significant therapeutic effect. Thus, the therapeutic effect of CAR-T therapy was related to the target antigen expression level, and the further retrospective analysis of pathological findings from PDAC patients showed a correlation between the intensity of CEA immunostaining and tumor heterogeneity. Therefore, CEA expression levels in biopsies or surgical specimens can be clinically used as biomarkers to select PDAC patients for anti-CAR-T therapy.

## 1. Introduction

Pancreatic ductal adenocarcinoma (PDAC) is one of the most lethal malignancies. Despite substantial improvements in the survival rates for other major cancer types, the survival rates for patients with PDAC have remained relatively unchanged since the 1960s [1]. PDAC is usually detected at an advanced stage and most treatment regimens are ineffective, contributing to the poor overall prognosis [2,3]. Therefore, there are vast unmet medical needs in patients with PDAC.

Immunotherapies, such as immune cell therapy and immune checkpoint inhibitor immunotherapy, have revolutionized cancer treatment in recent years. Above all, chimeric antigen receptor T cell (CAR-T) therapy, which uses genetic engineering techniques, has gained attention as a form of immune cell therapy. More recently, CAR-Ts were developed using various signal transduction domains or intracellular domains to improve activity, and these developments have enhanced the cytotoxicity of CAR-Ts to cancer cells and safety [4,5,6,7]. There is a high therapeutic efficacy in blood cancers [7,8,9,10,11], but CAR-T therapy has not yet demonstrated satisfactory results in solid tumors.

In this study, we hypothesized that carcinoembryonic antigen (CEA) is a key target antigen for CAR-T therapy in PDAC, an intractable disease, because it is highly expressed on the surfaces of most PDAC cells [12]. We used CAR-T comprising a CEA-specific single-chain variable fragment (scFv), CD3ζ as the signaling domain, and GITR, which is a signaling domain that confers resistance to CD4^+^ regulatory T cells but activates CD8^+^ T cells, for examination. Previous reports have suggested that the efficacy and safety of CEA-CAR-T are useful [13,14,15], but have not yet been clinically applied. One of the factors affecting the therapeutic effects of CAR-T therapy for solid cancers is the existence of a tumor microenvironment and the trafficking of T cells to the tumor; thus, in vivo experiments in a more clinically similar environment were considered necessary. Therefore, we examined the antitumor effects of adoptive anti-CEA-CAR-T transfer in various CEA-positive human PDACs with different antigen expression levels, using orthotopic xenograft mouse models of PDAC. In addition, while the development of novel CAR-Ts recently improve therapeutic efficacy and safety [6], we consider that we consider that the selection of cases is an important factor as well as the development of novel CAR-T itself. Therefore, we further retrospectively performed immunostaining for CEA in surgical specimens obtained from PDAC patients, and suggested a criterion for selecting patients with PDAC who are likely to benefit from CAR-T therapy. The results of this study could contribute to the clinical realization of CAR-T therapy for PDAC.

## 2. Materials and Methods

### 2.1. Animals

NOD/Shi-scid, IL-2RγKO (NOG) mice, obtained from Japan In-Vivo Science Inc., were fed a standard diet, housed under specific pathogen-free conditions, and used when they were 8–10 weeks old. All mice were female. All animal experiments were conducted according to protocols approved by the Animal Care and Use Committee of Hokkaido University Institute Genetic Medicine.

### 2.2. Cell lines

The human PDAC cell lines MIA PaCa-2 (DMEM + 10% FBS + 2.5% horse serum), PANC-1 (DMEM + 10% FBS), Capan-1 (IMD + 20% FBS), and AsPC-1 (RPMI-1640 + 10% FBS), and PK-9 were purchased from the American Type Culture Collection (ATCC, Rockville, MD, USA). Other human PDAC cell lines, including SUIT-2 (EMEM + 10% FBS), KP1N (RPMI-1640 + 10% FBS), PK-9 (RPMI-1640 + 10% FBS) and human gastric cancer cell line MKN45 (RPMI-1640 + 10% FBS), were purchased from the Japanese Collection of Research Bioresources (JCRB, Osaka, Japan). The human PDAC cell line PCI-66 (RPMI-1640 + 10% FBS) was established and provided by Dr. H. Ishikura (the First Department of Pathology, Hokkaido University, Japan). Additionally, the luciferase stably expressing human PDAC cell line BxPC-3 (RPMI-1640 + 10% FBS) was purchased from JCRB. The base media for each cell line are selected following data sheets issued by suppliers in initial propagation. After maintaining stable growth, all cell lines were become familiar with RPMI 1640 medium supplemented with 10% FBS because of having used there for such a long time. 

For transfection of luciferase to PANC-1 and PK-9, the vector co-expressing green fluorescence protein and luciferase was individually introduced into Panc-1 and PK-9 using FuGENE 6 transfection reagent (#E2691; Promega, WI, USA), according to the manufacturer’s protocol. Monoclonal cells were established by limiting the dilution method with hygromycin. The vector was prepared by introducing the *GFP* gene into the pGL4.50[luc2P/CMV/Hygro] vector (#E1310, Promega). 

### 2.3. Analysis of CEA Expression on the Surface

CEA expression on cell surfaces in all cell lines was analyzed by immunofluorescence (IF) assays and flow cytometry. Mouse anti-CEA immunoglobulin G (IgG) primary antibody (ab105364; Abcam plc., Cambridge, UK) and Alexa Fluor 488 conjugated anti-mouse IgG secondary antibody (ab150113; Abcam plc.) were used to detect CEA expression on cell surfaces in all cell lines. Mouse IgG antibody (ab37355; Abcam plc.) was used as the isotype control antibody under the same conditions. For IF assays, cells were cultured on glass coverslips and incubated at 37 °C for 24 h, fixed with 4% PFA, and then blocked with 1% bovine serum albumin (BSA). Primary antibodies (1:100) were incubated overnight at 4 °C. The glass coverslips were washed with phosphate buffered saline (PBS), incubated with a secondary antibody (1:100) for 1 h at 4 °C in the dark and sealed with DAPI-Fluoromount-G (# 0100–20; SouthernBiotech, Birmingham, AL). The samples were examined under a fluorescence microscope (KEYENCE Co., Tokyo, Japan). For flow cytometry, the cells were first detached from the plate using trypsin, followed by blocking with 1% BSA, and incubation of samples with primary antibodies (10 µg for 10^6^ cells) for 1 h at 20 °C. The samples were washed with PBS, and then incubated with secondary antibody (1:2000) for 30 min at 4 °C in the dark. The number of CEA molecules on the cell surface of each cell line was calculated from mean fluorescence intensity (MFI) according to the procedure recommended by QIFIKIT (K0078; Dako Cytomation, Glostrup, Denmark) (Appendix A).

### 2.4. Western Blot Analysis

To extract total proteins, harvested cells were lysed and scraped in RIPA buffer supplemented with 1× Protease Inhibitor Cocktail (G6521; Promega), followed by sonication at 4 °C to disrupt the cells. A total of 20 µg protein extract was electrophoresed on 15% sodium dodecyl sulfate-polyacrylamide gel electrophoresis (SDS-PAGE) and then transferred to nitrocellulose membranes, which were blocked with 5% non-fat milk and incubated with the following specific mouse anti-human IgG primary antibodies for 1 h at 20 °C: CEA (ab105364; 1:100; Abcam plc.), and β-actin (ab8227; 1:5000; Abcam plc.). All samples were washed with Tris-buffered saline and incubated with horse radish peroxidase (HRP)-conjugated bovine anti-mouse IgG antibodies (#115–035–003; 1:10,000; Jackson ImmunoResearch Laboratories, West Grove, PA, USA) for 1 h at 20 °C. The results were visualized using a chemiluminescence detection system (Bio-Rad Laboratories, Hercules, CA, USA). 

### 2.5. Enzyme Linked Immunosorbent Assay (ELISA)

A commercial ELISA kit (ab183365; Abcam plc.) was used for the quantitative measurement of CEA in cell culture supernatant and mouse plasma, and another ELISA kit (ab174443; Abcam plc.) was used for the quantitative measurement of interferon (IFN)-γ in cell co-culture supernatant, according to the manufacturer’s protocols. Absorbance was measured at 450 nm using an ELISA reader. 

### 2.6. Flow Cytometry

Flow cytometry analysis was performed to detect CEA expression on cell surfaces in all cell lines, CAR-transduction rates of T cells, and cytotoxicity. These assays were performed on a MACS Quant Analyzer 10 (Miltenyi Biotec, Bergisch Gladbach, Germany) and analyzed with FlowJo 7.6.5 (TreeStar Inc., Ashland, OR, USA). 

### 2.7. Vector Construction and Preparation of Virus Solutions

The scFv of the monoclonal antibody F11-39 specific to CEA in the VL-VH orientation [16], along with a CD8a hinge, CD28 transmembrane domain and CD3ζ signaling domains, and GITR intracellular domain, were cloned into a pMS3 retroviral vector [17]. GITR plays a functional role in regulating the activity of regulatory T cells through GITR ligands [18]. The murine stem cell virus LTR was used to drive CAR expression. Ecotropic pseudotyped retroviruses were transiently obtained by conventional methods using 293T cells (ATCC CRL-3216) and Retrovirus Packaging Kit Eco (#6160; Takara Bio, Shiga, Japan). The PG13 cells (ATCC CRL-10686) were transduced with the ecotropic retroviruses to produce GaLV-pseudotyped retroviruses.

### 2.8. CAR-T Cell Production

Peripheral blood (10 mL) was collected from healthy donor and peripheral blood mononuclear cells (PBMCs) were isolated using Ficoll density centrifugation. The cells were expanded in culture medium for three days in a well pre-coated with anti-CD3 antibody (#16–0037–85; Invitrogen, Carlsbad, CA, USA) and RetroNectin (T202; Takara Bio). The culture medium was GT-T551 (WK551; Takara Bio) supplemented with recombinant IL-2 (Novartis, Emeryville, CA, USA) at 600 IU/µL and autologous plasma. On the third and fourth days after the stimulation (on days 3 and 4), 2 × 10^5^ cells were transduced with the viral vector using the RetroNectin-bound virus infection method, wherein virus solutions were preloaded onto RetroNectin-coated wells in a 24-well plate. On day 5, the cells were transferred to a 50 mL flask containing culture medium for expansion. On days 10–14, cells were harvested and used for experiments. To confirm CAR-transduction of CD4^+^/CD8^+^ T cells, the following antibodies were used for cell surface staining by flow cytometry: APC-conjugated anti-CD4 (#300552; 1:50; BioLegend, San Diego, CA, USA), PE-Cy5 conjugated anti-CD8 (#555636; 1:50; Biosciences, San Jose, CA, USA), PE-conjugated streptavidin (#130–106–790; 1:300; Miltenyi Biotec), and biotinylated recombinant CEA (0.1 µg for 2 × 10^5^ cells): recombinant CEA (ab158095; Abcam plc., Cambridge, UK) was biotinylated using the Biotin Labeling Kit (Dojindo, Kumamoto, Japan) according to the manufacturer’s protocol.

### 2.9. CAR-T Cell Sorting

The prepared lymphocyte solution containing CAR-T cells was incubated with biotinylated recombinant CEA (1 µg for 1 × 10^7^ cells) for 30 min at 20 °C. Cells were washed with buffer and incubated with anti-biotin microbeads (20 µg for 1 × 10^7^ cells; Miltenyi Biotec) for 15 min at 4 °C. After washing the cells, CAR-T cells were isolated using magnetic cell separators (Miltenyi Biotec) in a dedicated column. Separation efficiency was evaluated by flow cytometry, and it was confirmed that the sorting was sufficient (Appendix A).

### 2.10. Cytotoxicity Assay

Cytotoxicity of CEA-CAR-T cells against PDAC cells was analyzed by flow cytometry using the 7-AAD/CFSE Cell-Mediated Cytotoxicity Assay Kit (#600120; Cayman Chemical, MI, USA). All PDAC cell lines and MKN-45 cells were labeled with CFSE for 10 min at 37 °C. Stained cells were washed twice with PBS and mixed with CEA-CAR-T cells at effector-to-target ratios (E:T) of 1:1 and 10:1, followed by 6 h of incubation at 37 °C. After washing with PBS, the cell mixtures were incubated with 7-AAD for 15 min and analyzed by flow cytometry. CFSE^+^ target cells were gated and 7-AAD exclusion was performed to visualize the proportions of live and dead cells percentages. The assay was also performed in the presence of soluble CEA at 1000 ng/mL and T cells without CAR were used as the control.

### 2.11. Orthotopic Pancreatic Ductal Adenocarcinoma Xenograft Models

Luc expressing human PDAC cells were induced in NOG mice via intrapancreatic injection of 1 × 10^6^ PANC-1 (CEA^−^), PK-9 (CEA^+^), and BxPC3 (CEA^++^) cells. In vivo imaging was performed after intraperitoneal injection of D-luciferin (#LK10000; 3 mg/mouse; Funakoshi Co., Tokyo, Japan) as a substrate for Luc, then visualized and analyzed using the IVIS Spectrum Imaging System (PerkinElmer Inc., Waltham, MA, USA) for tumor growth slopes. Gating of the regions of interest was unified, covering the entire abdomen. Photon emission intensity (photons/sec/cm^2^/sr) was calculated based on the data of photons emitted from the respective regions of interest using Living Image software (PerkinElmer Inc., Boston, MA, USA).

### 2.12. In Vivo CAR-T Cell Treatment

Orthotopic xenograft models were randomly assigned to the treatment and no-treatment groups (*n* = 3–6 for all groups). Orthotopic mouse models were prepared on day 7 and therapeutic intervention was administered on day 0. Mice in the treatment group were injected with 2.5 × 10^6^ CAR-T cells in 200 µL PBS via the tail, and those in the no-treatment group were injected with 2.5 × 10^6^ NGMC cells in 200 µL PBS. Tumor growth was monitored once a week using IVIS imaging. Changes in weight over time as well as the levels of blood CEA and cytokines were measured to evaluate the side effects of the treatment. Peripheral blood samples were collected from mice and separated into serum and plasma. Serum CEA level was measured by ELISA, while plasma was analyzed using cytokine bead arrays (Bio-Plex, Multiplex Immunoassay, Bio-Rad Laboratories, Hercules, CA, USA). Mice were euthanized for humane reasons on day 21. This result was the summary of several experiments under the same conditions.

### 2.13. Patients

Table 1 shows the characteristics of patients with PDAC in the retrospective study. A total of 151 patients who underwent surgery at Hokkaido University Hospital from 2009 to 2016 were included in this study. The cutoff threshold of serum CEA level at one month before surgery was arbitrarily set as 5 ng/mL and defined as the cutoff threshold in our premises. We randomly selected 22 cases and performed tissue microarray (TMA) of resected specimens for histopathological examination.

### 2.14. Immunohistochemical Staining 

Paraffin-embedded sections of mouse tissues fixed with 10 % buffered formalin were subjected to immunohistochemical staining by the HRP-labeled polymer method (EnVision FLEX system, Dako, Santa Clara, CA, USA). Mouse monoclonal anti-CEA (M7072; Dako), mouse monoclonal anti-CD8 (IR623; Dako), and mouse monoclonal anti-CD4 (IR649; Dako) antibodies were used as primary antibodies and reacted with 3,3′-Diaminobenzidine (DAB) chromogenic substrate. The sections were stained with hematoxylin and eosin (H&E). Tissue immunostaining for CEA was performed on TMA prepared from PDAC specimens collected after surgery, and the staining intensity and heterogeneity were evaluated. By only focusing on the staining intensity of the cell membrane surfaces, the intensity was divided into three grades, based on the degree of staining that occupies the largest area of the cancer cells, as follows: CEA^−^, CEA^+^, and CEA^++^. In addition, to evaluate heterogeneity, the area ratio of CEA-positive cells to all tumors in each TMA section was divided into six groups: 0%, 0–25%, 25–50%, 50–75%, and 75–100%. Each area was evaluated under a microscope (KEYENCE Co.). 

### 2.15. Statistical Analysis

Data are presented as median ± SEM, where error bars are shown. Statistical analysis was performed using JMP pro13 (SAS Institute Inc., Cary, NC, USA). Two-tailed unpaired Student’s *t*-tests were used to determine significant differences in enumeration assays. Correlative analyses were performed using Pearson’ s correlation coefficient (*R*). The Kruskal-Wallis test was used to compare differences between three or more unpaired groups. Statistically significant differences between variables in the contingency table were analyzed using Fisher’s exact test. Differences were defined as statistically significant if the *p* value was < 0.05, while *R* > 0.7 was defined as a strong correlation.

## 3. Results

### 3.1. CEA Expression on Human Pancreatic Cells

CAR-T recognize the target antigen of tumor cells directly without the major histocompatibility complex (MHC), so molecules on the surface of the cell membrane are the target antigens. To evaluate CEA expression on cell membranes, 10 PDAC cell lines were examined by IF, Western blot (WB) analysis, and flow cytometry (Figure 1A–C). In flow cytometry, the number of CEA molecules on the cell surface was evaluated based on the MFI of CEA positive by using the calibration curve. In the evaluation of cellular CEA by WB, CEA protein was detected higher than the result of FCM in some cell lines. We speculate that this discrepancy was an overestimation of CEA as the target antigen because it was detected not only on the cell membrane but also in the cell endogenous CEA protein by WB. Therefore, referring the number of CEA molecules, BxPC-3 and MKN45 were identified as cell lines with high CEA expression levels (CEA^++^), while Capan-1, AsPC-1, and PK-9 were identified as cell lines with medium CEA expression levels (CEA^+^), and MIA Paca-2, Panc-1, SUIT-2, KP1N, and PCI-66 were categorized as cell lines with low CEA expression levels (CEA^−^). 

### 3.2. Soluble CEA Secretion by Pancreatic Cell Lines In Vitro

CEA is anchored to cell membrane by a glycosyl-phosphatidylinositol moiety, and it is cleaved with phosphatidylinositol-specific phospholipase C to become soluble CEA [19]. To determine whether the presence of soluble CEA interferes with anti-CEA-CAR-T function, the relationship between CEA expression on cell membranes and soluble CEA secretion by cells in each cell line was examined. Soluble CEA secreted in the culture medium was measured by ELISA after 72 h of seeding 1× 10^6^ cells (1 mL) into a 10 cm dish, and the median CEA concentrations for the different cell lines were lower than the sensitivity limit in cases of all CEA negative cell lines (Figure 2). The concentrations of soluble CEA in the culture media strongly correlated with the number of CEA molecules (correlation coefficient *R* = 0.668, *p* = 0.102). 

### 3.3. ELISA Revealed Anti-CEA-CAR-T Activation against CEA Positive Cells

To verify the antitumor effect of CAR-T on CEA positive PDAC cells, we analyzed modified T cells incubated with PDAC cell lines and, as a control, with MKN45. CAR non-introduced T cells (None Gene Modified Cells: NGMC) were used as the control group. ELISA revealed that IFN-γ secretion was significantly increased in the co-cultures of CAR-Ts with CAPAN-1, AsPC-1, BxPC-3, PK-9, and MKN45, which expressed CEA at high and moderate levels (CEA^++^ and CEA^+^). There was no statistically significant difference among co-cultures with CEA^−^ cell lines (Figure 3A,B). Notably, the IFN-γ secretion levels were correlated with the number of CEA molecules (correlation coefficient *R* = 0.653, *p* ≤ 0.100).

### 3.4. Cytotoxicity Assay Showed Anti-CEA-CAR-T Activation against CEA Positive Cells

In the cytotoxicity assay using the 7-AAD/CFSE assay kit, which had an effector: target ratio of 10:1, the median cytotoxicity values for the different cell lines in the CAR-T incubation and the NGMC groups result that the CEA^++^ cells BxPC-3, Capan-1, and MKN45 showed significant dose-dependent cytotoxicity, whereas the CEA^+^ cells AsPC-1 and PK-9 showed slight cytotoxicity but no significant. No difference in cytotoxicity was observed between the CAR-T and the NGMC assays of other cell lines, which had almost no CEA expression on the cell membrane (Figure 4A). Notably, the cell lines BxPC-3 and MKN45, which showed significant cytotoxicity, had the highest number of CEA molecules. Besides, in the CEA competition assay, different concentrations of recombinant soluble CEA were added to the co-cultures (Figure 4A,B). In particular, IFN-γ secretion was not significantly affected by soluble CEA concentrations of 10 ng/mL, 100 ng/mL, and 1000 ng/mL, compared to that when no soluble CEA was added. On the other hand, a high concentration of soluble CEA (1000 ng/mL) resulted in the suppression of cytotoxicity of CAR-T for BxPC-3 and MKN45 significantly.

### 3.5. Adaptive Therapy with Anti-CEA-CAR-T Caused Differential Antigen Expression on Orthotopic Xenograft Models of CEA Positive PDAC

Orthotopic xenograft models were created from NOG mice using human PDAC cell lines, and experimental treatments were administered. To examine the difference in the antitumor effects of CAR-T due to differences in CEA expression levels between cell lines in vivo, which was confirmed in vitro, luciferase was introduced into CEA++ BxpC3, CEA+ PK-9, and CEA- PANC-1 as a reporter gene via injection into mouse pancreas. The orthotopic mouse models were established on day 0 (7 days after the transplantation of cell lines), and CAR-Ts were adoptively transferred via injection into the tail vein. NGMC was injected as a control group. After a single administration of CAR-T, the enhancement of tumor luciferase signal was significantly decreased in the BxPC-3 model on days 14 and 21, indicating significant suppression of CEA^++^ pancreatic cancer growth due to CAR-T therapy. On the other hand, there was no significant suppression in the other two models (Figure 5A). In addition, to evaluate the side effects of CAR-T administration, changes in body weight over time and levels of inflammatory cytokines in the blood were analyzed (Figure 5B,C). There was no treatment-related difference in body weight in any of the mouse models. Serum levels of cytokines were evaluated by measuring the levels of IFN-γ, IL-6, and TNFα in blood on day 21. Changes in serum levels of cytokines between the CAR-T and NGMC groups were similar in all mouse models, with no significant difference; however, cytokine release tended to be higher in the CAR-T group in all models.

### 3.6. Immunohistochemical Evaluation of Transplanted Tumor following Anti-CEA-CAR-T Treatment

Mice were sacrificed on day 21 (28 days after tumor injection) and the tumors were removed for analysis by immunohistological staining (Figure 6A). Staining for CEA in BxPC-3 tumors was consistent in almost all tumor cells. Moreover, in the CAR-T group, CD8^+^ T-cells were clustered together in the tumor. In PK-9 tumors, almost no CEA staining was observed, although CD8^+^ T-cells clustered toward the tumors. The PANC-1 tumors did not show any CEA staining, although some individuals in the NGMC group showed clustering of CD8^+^ T-cells toward the tumor. To examine the therapeutic effects of CAR-T, the tumor areas were divided into center and peritumor, and CD8^+^ T-cells were counted within the range of 0.25 mm^2^/field (Figure 6B). The infiltration of CD8^+^ T-cells was higher in the centers than in the peritumor areas of BxPC-3 and PK-9 tumors in the CAR-T group but clustered at the centers and peritumor areas in BxPC-3 tumors of the NGMC group. In PANC-1 tumors of the CAR-T group, almost no infiltration of CD8^+^ T-cells was observed, while slight infiltration was observed in those of the NGMC group. 

### 3.7. Soluble CEA Did Not Markedly Block Anti-CEA-CAR-T Activation

CEA is soluble and present in the serum of healthy individuals. In the in vitro experiment, cytotoxic activities of CAR-T against some cell lines were suppressed under high concentrations of CEA (1000 ng/mL), but no significant difference in the cytotoxicity of CAR-T was noted at CEA concentrations < 100 ng/mL (Figure 3A,B). In the in vivo experiment, serum CEA levels measured before (on day 0) and after (on day 21) treatment, serum CEA levels before the treatment were lower in all cell lines, regardless of whether they were CEA positive or negative cells (Figure 7). In other words, soluble CEA did not interfere with CAR-T function in the orthotopic xenograft mouse models. On the contrary, serum CEA levels were significantly increased in the BxPC-3 model of the CAR-T group but remained unchanged in that of the NGMC group after treatment. 

### 3.8. Identification of Surrogate Markers by Retrospective Analysis of PDAC Cases to Select PDAC Patients for Anti-CEA-CAR-T Therapy 

Anti-CEA-CAR-T showed antitumor effects on CEA positive cells, suggesting that it might be more effective in tumors with less heterogeneity both in vitro and in vivo. We examined the indices to select PDAC patients, who could benefit from the therapeutic effect of CAR-T, for potential clinical applications. The retrospective analyses of clinical data and pathological findings of CEA staining were performed for 151 patients with PDAC, including 89 males (58.9%) and 62 females (41.1%), who underwent surgery at Hokkaido University Hospital from 2009 to 2016. Table 1 shows the patient characteristics; the median age was 68 years, serum CEA level was 4.2 ng/mL, and CEA positive rate was 36.4% (55/151). Of these patients, 22 were randomly selected for TMA. The selected patients included 12 men (54.5%) and 10 women (45.5%), with a median age of 68.5 years, serum CEA level of 3.95 ng/mL, and CEA positive rate of 36.4%. It was confirmed that there was no bias in selecting patients for TMA. Anti-CEA staining was performed with TMA, and the specimens were categorized into three grades based on staining intensity and five grades based on heterogeneity (Figure 8A,B). There was no significant correlation between preoperative serum CEA levels and CEA staining intensity (Figure 8C). Table 2 shows the results of the analysis for correlation between heterogeneity and serum CEA positivity or CEA staining intensity of cell membranes, which indicate that heterogeneity and serum CEA positivity were not correlated but heterogeneity and staining intensity were significantly correlated (*p* < 0.001). 

## 4. Discussion

CAR-T therapy has shown remarkable therapeutic effects in patients with blood tumors such as leukemia or lymphoma, and CTL019 was approved by the FDA in 2017. However, CAR-T therapy has lesser remarkable therapeutic effects in solid tumors than in blood tumors, and various studies are being conducted to overcome its limitations [4,20]. Recent pilot studies [21,22,23,24] with CAR-T targeting various antibodies which are specific to solid tumors, such as ERBB2, PSMA, or mesothelin, showed inadequate therapeutic efficacy. Hence, the factors influencing the therapeutic efficacy of CAR-T are pathological and immunological, such as the heterogeneity of target antigens, the microenvironments of solid tumors, and the trafficking of CAR-T to tumors [25,26,27,28,29,30]. In this study, we focused on PDAC, which is one of the most lethal malignancies. Despite substantial improvements in the survival rates for other major cancer types, the survival rates for PDAC have remained relatively unchanged since the 1960s [1]. PDAC is usually detected at an advanced stage and most treatment regimens are ineffective, contributing to the poor overall prognosis [2,3]. Therefore, PDAC is one of the diseases with vast unmet medical needs. Notably, we focused on CEA, which is highly expressed in pancreatic cancer, as the target antigen, and analyzed the efficacy of anti-CEA-CAR-T therapy in PDAC based on CEA expression levels [12]. We also verified the safety and efficacy of CAR-T for treating PDAC using an orthotopic mouse model transplanted with human PDAC cell lines. These models represent the entire process of the metastasis, consisting of local tumor growth, vascular and lymphatic invasion at the local site, flow in the vessels and lymphatic, extravasation at the metastatic organs, and seeding and growth at relevant metastatic sites, similar to that of clinical cases.

First, direct, and indirect functional assays were performed to evaluate the effects of anti-CEA-CAR-T on various PDAC cell lines in vitro, and determine the relationship between these therapeutic effects and CEA expression on cell membranes. In vitro functional assays of CAR-T for the different cell lines revealed that the stronger the antigen expression, the more significant the antitumor response of CAR-T. Subsequently, we assessed the therapeutic effects of anti-CEA-CAR-T in vivo. To do this, one cell line was representatively selected from each of the high, medium, and low CEA expression of cell membranes groups, and orthotopically transplanted into NOG mice to create orthotopic xenograft models. The orthotopic models with BxPC-3 as the CEA^++^ cells showed suppression of tumor growth on days 14 and 21 after treatment with a single intravenous dose of CAR-T. Interestingly, T cells without CAR did not affect tumor growth, suggesting that the antitumor effect was mediated specifically by CAR. On the contrary, antitumor activity of CAR-T was not observed in the PK-9 and PANC-1 models, since the specimens of these pancreatic orthotopic models did not show pathological staining for CEA expression. The threshold antigen expression level for achieving antitumor effect such as HER2 or mesothelin for CAR-T therapy of solid tumors is defined, and therapeutic experiments have previously been conducted in this regard [22,30]; however, there are no reports on the correlation between antitumor effect and antigen expression level. Although previous studies on anti-CEA-CAR-T [14,31,32] have not confirmed the effect of different antigen expression levels on the antitumor effect, in vitro results of the present study showed that the therapeutic effect of CAR-T therapy may depend on the antigen expression level. In pathological findings, the number of CD8^+^ T-cells infiltration into tumors depending on the antigen expression level did not correlate with antitumor effects. For considering how the functionality of the infiltrated CAR-T improved, further staining of the actual function of each infiltrating lymphocyte with a cytotoxic substance such as perforin or granzyme may allow better assessment of functional changes of clustered CD8^+^ T-cells depending on the antigen expression level within the tumor environment. Since soluble CEA present in the serum of cancer patients blocks the CAR receptor [33,34], we examined whether its presence interfered with the cytotoxic activity of CAR-T. Although CEA concentrations as high as 1000 ng/mL suppressed cytotoxic activities in some cell lines in vitro, CEA concentrations up to 100 ng/mL did not significantly impact CAR-T activity, possibly because CEA is membrane-bound in preformed microdomains and thus, more likely to result in CAR clustering and the formation of a functional synapse than its mono- or dimeric forms in solution. In vivo, serum CEA levels in mice did not differ among the different orthotopic mouse models at the time of treatment intervention. Furthermore, since serum CEA level was <100 ng/mL, the antitumor response of CAR-T was not affected. On the contrary, serum CEA levels increased significantly in the CEA^++^ models after successful treatment, indicating the release of intracellular CEA resulting from the disruption of CEA^++^ tumor cells due to the action of CAR-T. A previous study [35] has reported that, in clinical practice, the median serum CEA level in PDAC patients is 3.51 ± 2.34 ng/mL, and in this retrospective study of patient data, the preoperative median serum CEA level was 4.2 ng/mL, which was similar to that in the previous report. Therefore, we do not expect a major impact of soluble CEA on the CAR redirected T-cell anti-tumor response in clinical applications.

Second, we examined the safety of anti-CEA-CAR-T therapy. Cytokine release syndrome (CRS) have been reported as side effects of anti-CEA-CAR-T therapy [36]. Therefore, we measured the serum levels of IFN-γ, IL-6, and TNFα in mice as representative biomarkers for predicting CRS severity, but found no significant difference in cytokine secretion irrespective of an antitumor effect. In all models, the release of cytokines tended to be higher in the CAR-T treatment group than in the NGMC group, indicating the CAR-T response to CEA expression on cell membranes or the non-specific CAR-T response to antigen-negative cells. Although it is unclear whether the release of these cytokines causes adverse events in mice, but at least there was no significant weight loss as seen in previous report [14]. Nevertheless, there are certain limitations to this safety result. On the one hand, one death was confirmed in the CAR-T treatment group for BxPC-3, while each of the CAR-T and NGMC treatment groups for PANC-1 had one death. We presume that the cause of death was likely associated with the invasive procedure of tumor transplantation; however, pathological analysis was not performed and, therefore, the possibility of side effects due to CAR-T cannot be completely ruled out.

Finally, for the potential clinical application of anti-CEA-CAR-T, we searched for biomarkers that can be used to select patients who are likely to benefit from the treatment. As mentioned previously, the serum CEA concentration cannot be regarded as a surrogate marker because it showed no correlation with CEA staining of speciCD3mens from PDAC patients. In contrast, the intensity of CEA immunostaining of pathological specimens from PDAC patients was correlated with the evaluation of CEA heterogeneity. The presence of heterogeneity at histological and molecular levels in tumors of PDAC were analyzed [37], which is one of the factors that make PDAC an intractable disease [38]. It was suggested that PDAC produces heterogeneous CEA molecules in tumors [39], but there are no systematic reports of CEA immunostaining heterogeneity. In this study, it is considered that clinical cases of PDAC with high CEA staining level have less heterogeneity. Hence, the CEA staining intensity of surgical or biopsy specimens from PDAC patients may be used to select target cases for CAR-T therapy. Furthermore, in recent years, the development of the detection of molecular abnormalities in liquid biopsies may eliminate the sampling bias caused by tumor heterogeneity [40,41], thus holding a very strong potential for novel approaches in the therapeutic management of cancer patients. 

## 5. Conclusions

In conclusion, adoptive cell therapy with anti-CEA-CAR-T induced the regression of CEA positive PDAC in orthotopic mouse models. More specifically, tumors with high CEA expression levels induced the infiltration of T cells into the tumors and increased therapeutic effects. By extension, the pathological findings of CEA expression level in one section from the specimen can be a biomarker in the selection of PDAC patient for clinical applications.

## Figures and Tables

**Figure 1 cancers-15-00601-f001:**
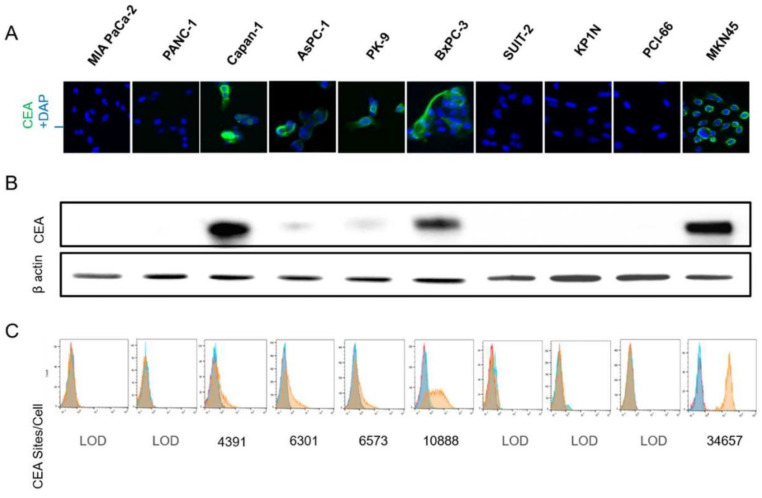
In vivo antigen quantification. (**A**) Immunofluorescence microscopy was used to detect CEA expression on cell membranes in cell lines. (**B**) Western blot analysis of whole-cell lysates using the indicated antibody detected the 180 kDa CEA, or anti-βactin antibody as the internal control. (**C**) Flow cytometry of each cell line tested qualitatively and quantitatively for CEA expression. For each graph, the red curve corresponds to cells, primary anti-CEA antibody, and secondary Alexa Fluor 488 conjugated antibody; the blue curve represents cells and isotype control.

**Figure 2 cancers-15-00601-f002:**
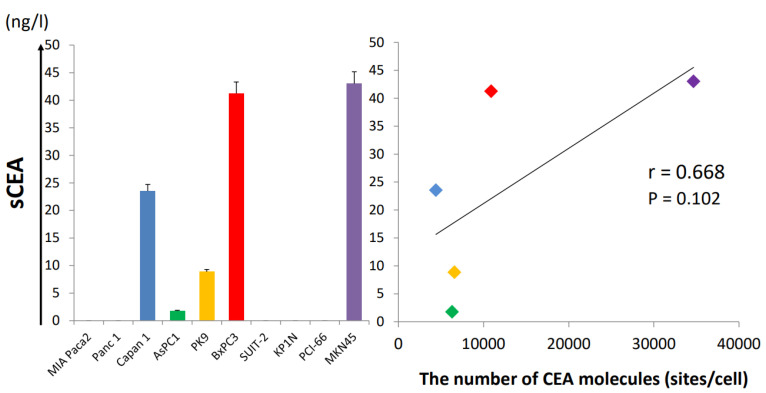
Soluble CEA concentration in cell culture media shows a correlation with the number of CEA molecules.

**Figure 3 cancers-15-00601-f003:**
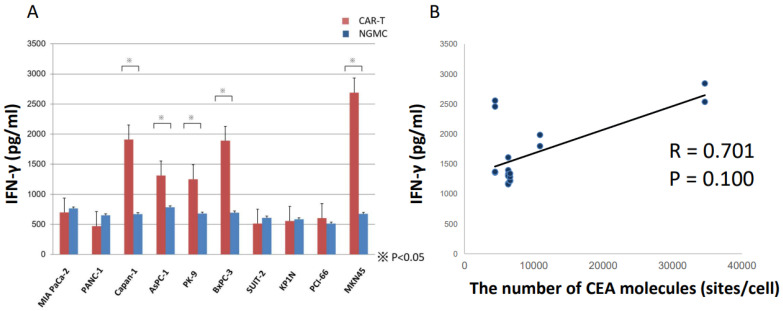
In vitro functional assays: ELISA. (**A**) Quantitative measurement of IFN-γ in cell supernatant after co-culture with CAR-T. (**B**) The secretion levels of IFN-γ were correlated with the number of CEA molecules. ^※^ *p* < 0.05.

**Figure 4 cancers-15-00601-f004:**
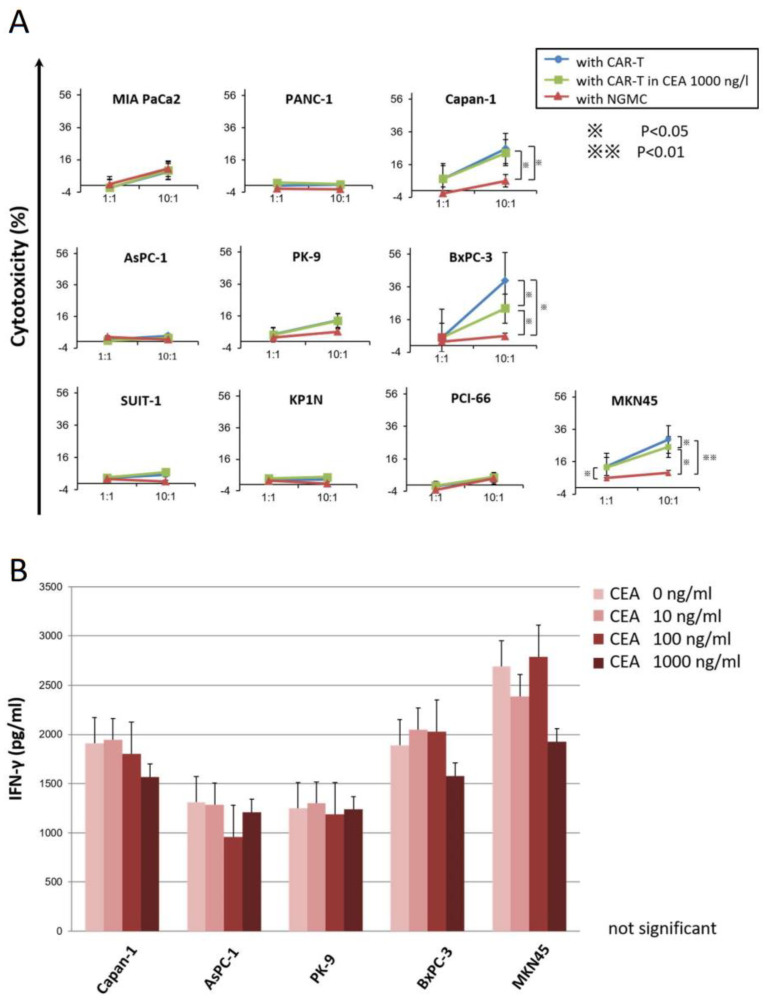
In vitro functional assays: Cytotoxicity assay. (**A**) Cytotoxicity assay with Effector: Target ratio of 10:1 in the presence of soluble CEA at concentrations of 0 ng/mL and 1000 ng/mL. The group without CAR-T was used as the control group. (**B**) CEA competition assay was performed by adding different recombinant CEA concentrations to the co-culture. ^※^ *p* < 0.05, ^※※^ *p* < 0.01.

**Figure 5 cancers-15-00601-f005:**
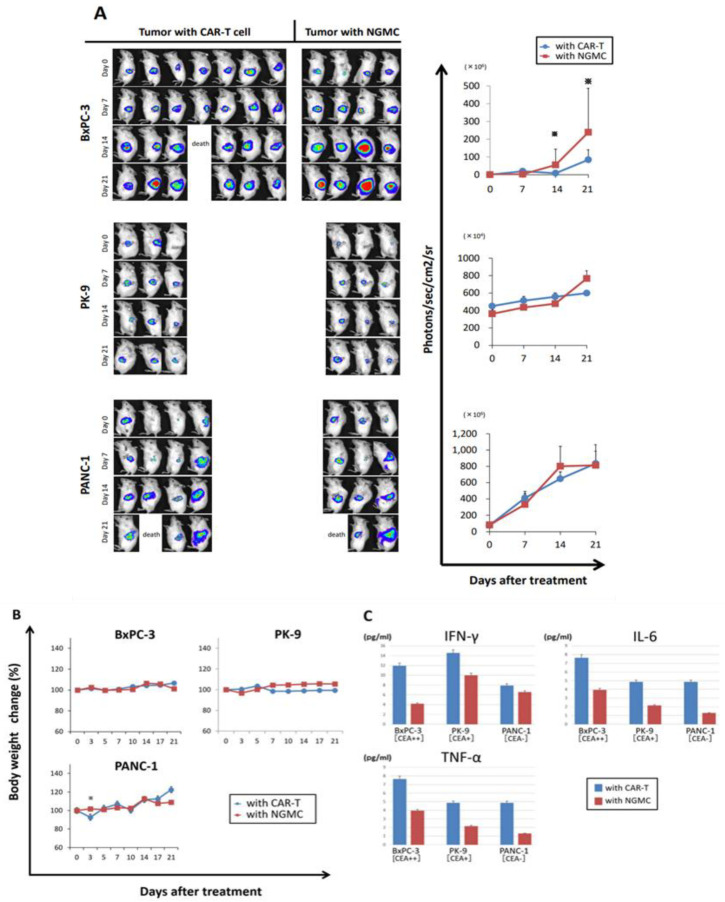
Suppression of transplanted pancreatic carcinoma upon adoptive therapy with anti-CEA-CAR-T. (**A**) Orthotopic PDAC mouse models were prepared by intrapancreatic injection of CEA^−^ (PANC-1), CEA^+^ (PK-9) or CEA^++^ (BxPC-3) pancreatic carcinoma cell lines (1× 10^6^ cells/mouse) in NOG mice, each type of cells was marked with luciferase for bioluminescence imaging. When tumors were established, anti-CEA-CAR-T was injected into the tail vein at day 0 (2.5×10^6^ cells/mouse). For comparison, T cells without CAR were injected. Tumor growth was monitored once a week using IVIS imaging. The enhancement of tumor luciferase signal was significantly decreased in only the BxPC-3 model on days 14 and 21. (**B**) Changes in weight over time. There were no significant weight changes in all groups. (**C**) Serum levels of cytokines in mice were measured. Pooled serum samples from mice were collected on day 21 and subjected to analysis by cytokine beads arrays. Although cytokine release tended to be higher in the CAR-T group in all models, there were no significant difference of changes in serum levels of cytokines between the CAR-T and NGMC groups.

**Figure 6 cancers-15-00601-f006:**
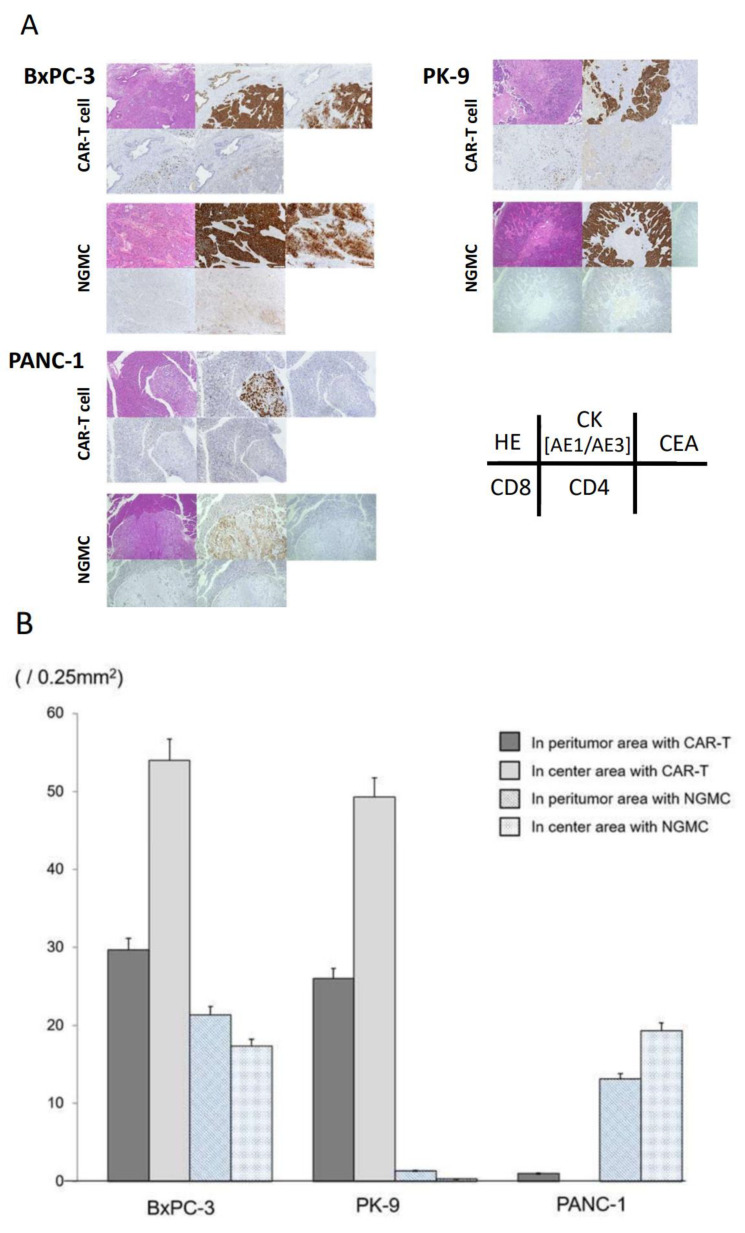
Histopathological examination of specimens collected from mice. (**A**) Tissues from orthotopic mouse models treated with anti-CEA-CAR-T or non-modified T cells on day 21 were stained with hematoxylin and eosin and analyzed for CD4, CD8, CK, and CEA expression by antibody staining. (**B**) Counts of CD8^+^ T cells infiltrating into the center and peritumor area within the range of 0.25 mm^2^/field.

**Figure 7 cancers-15-00601-f007:**
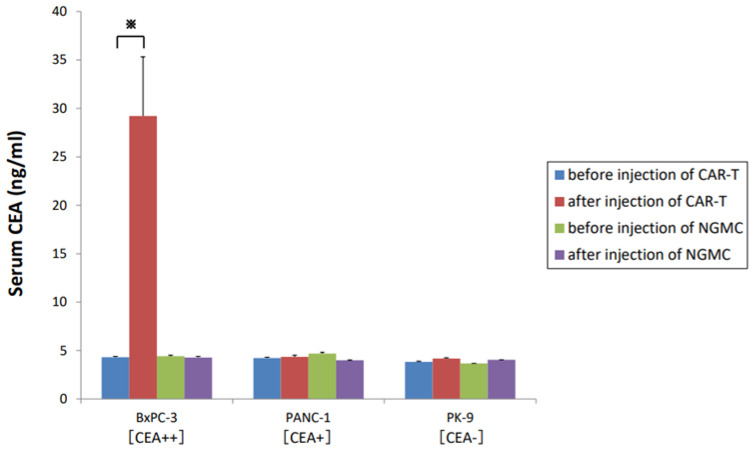
Serum level of CEA was measured by ELISA. Pooled serum samples from mice were collected before and after injection of T cells on days 0 and 21.

**Figure 8 cancers-15-00601-f008:**
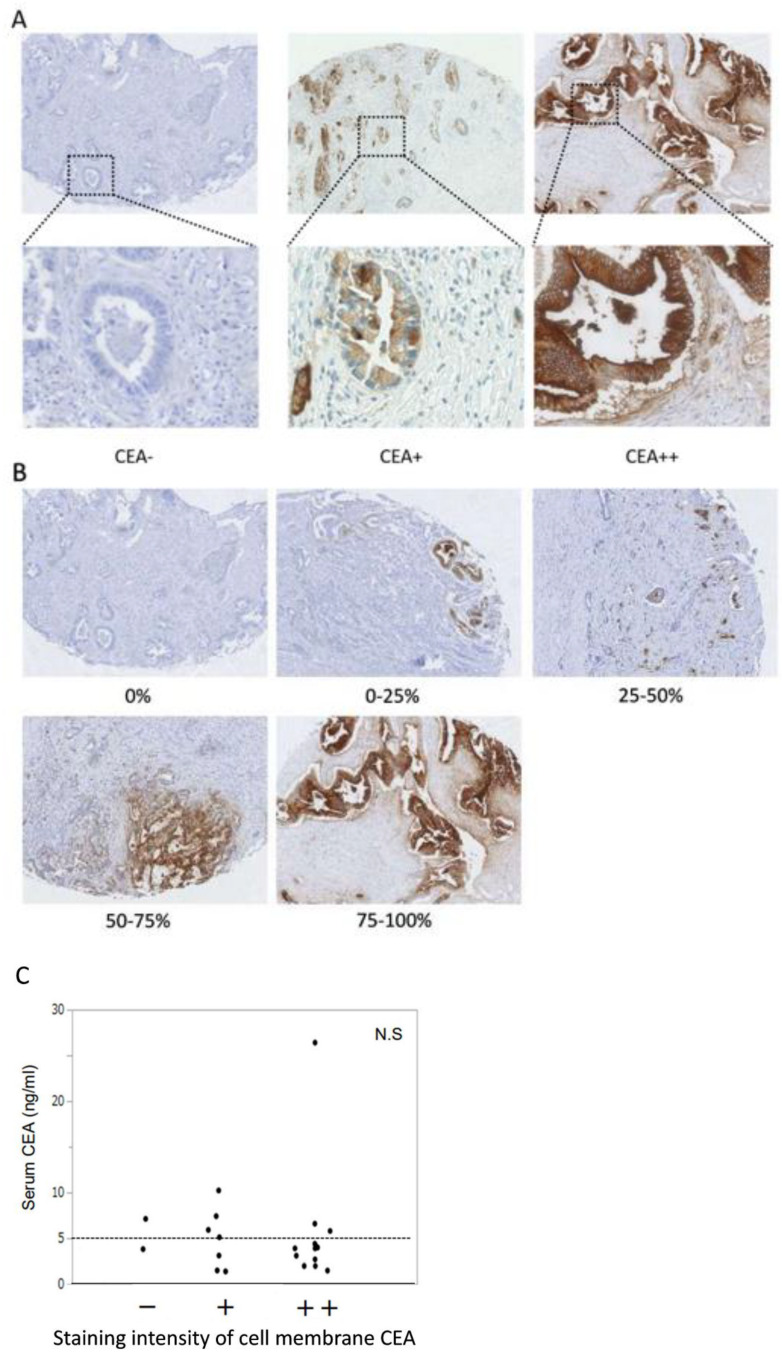
Immunohistochemical staining for CEA in PDAC patients. (**A**) CEA staining intensity was categorized into three grades. (**B**) Heterogeneity was classified into five grades. (**C**) Analysis of correlation between serum CEA and staining intensity of cell membrane CEA. There was no significant correlation.

**Table 1 cancers-15-00601-t001:** Characteristics of pancreatic adenocarcinoma patients.

**Characteristics of All Postoperative Patients**	
	***n* = 151**
Gender	
Male	89 (58.9)
Female	62 (41.1)
Age (years old)	68 (42–83)
Serum CEA (ng/mL)	4.2 (0.6–212)
<5	96 (63.6)
≥5	55 (36.4)
**Characteristics of randomly selected patients for Tissue Microarray**	
	***n* = 22**
Gender	
Male	12 (54.5)
Female	10 (45.5)
Age (years old)	68.5 (55–82)
Serum CEA (ng/mL)	3.95 (1.5–26.4)
<5	14 (63.6)
≥5	8 (36.4)

Characteristics of all postoperative pancreatic adenocarcinoma patients and randomly selected patients for Tissue Microarray. Data were presented as values are *n* (%) or median (range).

**Table 2 cancers-15-00601-t002:** The correlation of heterogeneity with serum CEA positivity or staining intensity of CEA.

	Heterogeneity (%)					*p* Value
	0	0–25	25–50	50–75	75–100	
Serum CEA (ng/mL)						0.731
<5	1	1	1	2	10	
≥5	1	0	1	0	5	
Intensity						<0.001 *
CEA^−^	2	0	0	0	0	
CEA^+^	0	1	2	2	2	
CEA^++^	0	0	0	0	13	

Values are *n*; *p* values were calculated using the Fisher’s exact test; * Significantly different.

## Data Availability

Data are contained within the article or Appendix A.

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
