# Peer review of "Tumor Growth Suppression of Pancreatic Cancer Orthotopic Xenograft Model by CEA-Targeting CAR-T Cells"

_cancers, 2023, doi:10.3390/cancers15030601_

Round 1

Reviewer 1 Report

In this manuscript, the authors conducted an experimental study to examined the effect of CAR-T therapy targeting carcinoembryonic antigen (CEA) in pancreatic adenocarcinoma (PDAC) model, and found that only the cell line with high CEA expression exhibited a significant therapeutic effect. Furthermore, the pathological findings from PDAC patients showed that tumor heterogeneity was correlated with the intensity of  CEA immunostaining, but not the serum CEA. So they concluded that CEA expression levels in biopsies or surgical specimens can be clinically used as biomarkers to select PDAC patients for anti-CAR-T therapy targeting CEA.

Overall, this study is of great interest and the manuscript is quite well-written. I have only some minor concerns about the current manuscript.

  1. The classification of high CEA expression was not clear. As shown in Figure1, Capan-1 were identified with high CEA expression levels by western blot and immunofluorescence microscopy. However, it was finally identified with medium CEA expression levels, perhaps due to the results of flow cytometry. Maybe the authors considered the CEA protein expressed on the membrane of cancer cells as more accurate therapeutic target than the endogenous CEA protein expressed inside the cells. But this was inconsistent with immunohistochemical staining, which was concluded as the potential biomarker detection of anti-CAR-T therapy targeting CEA.

  2. In Figure5A, why the numbers of animals in each group were not the same.

  3. The last sentence in Page 16 “This section may be divided by subheadings. It should provide a concise and pre- 419 cise description of the experimental results, their interpretation, as well as the experi- 420 mental conclusions that can be drawn” looks like a comment.

  4. In figure 8B, the last 2 panels lacked annotation.

Author Response

Response to Reviewer 1 Comments

1.

The classification of high CEA expression was not clear. As shown in Figure1, Capan-1 were identified with high CEA expression levels by western blot and immunofluorescence microscopy. However, it was finally identified with medium CEA expression levels, perhaps due to the results of flow cytometry. Maybe the authors considered the CEA protein expressed on the membrane of cancer cells as more accurate therapeutic target than the endogenous CEA protein expressed inside the cells. But this was inconsistent with immunohistochemical staining, which was concluded as the potential biomarker detection of anti-CAR-T therapy targeting CEA.

Response 1:

Thank you for this comment. As you say, we consider CEA on the cell membrane as a target antigen for anti-CEA-CAR-T. Therefore, we evaluated the immunohistochemical staining of patients by only focusing on the staining intensity of the cell membrane surfaces, regardless of the cytoplasmic staining. Please refer to Page 6, Lines 261-263.

I also added to the Result section and Figure 8 Legend. (Page 14, Line 431. Page 15; Line 444).

And we have now also added the sentence in the Results section to avoid any potential confusion: ' CAR-T recognize target antigen of tumor cells directly.…' (Page 7, Lines 279-281).

2.

In Figure5A, why the numbers of animals in each group were not the same.

Response 2:

Thank you for this comment.

Some mice died before injection of CAR-T, i.e., immediately after the surgery for creating orthotopic models, and some individuals did not grow tumors in pancreas. These technical failure cases were excluded beforehand, so the numbers in the cohort are uneven.

3.

The last sentence in Page 16 “This section may be divided by subheadings. It should provide a concise and precise description of the experimental results, their interpretation, as well as the experimental conclusions that can be drawn” looks like a comment.

Response 3:
We thank the reviewer for the careful review.

I’m very sorry, there was some unnecessary text left. I deleted it.

4.
In figure 8B, the last 2 panels lacked annotation.

Response 4:
We thank the reviewer for the careful review.

I’m very sorry, the annotations disappeared when placing the figure. I corrected it.

Reviewer 2 Report

In the current study titled “Successful tumor shrinkage of pancreatic cancer orthotopic xenograft model by CEA-targeting CAR-T cells”, the authors examined the therapeutic effect of CAR-T therapy targeting carcinoembryonic antigen (CEA) in 41 pancreatic adenocarcinomas (PDAC). In the study, the membranous expression of CEA was determined qualitatively and quantitatively by immunofluorescence, flow cytometry and western blotting and found that CEA is expressed significantly high in some PDAC cell lines compared to others. Authors have used 10 different PDAC cell lines for this purpose. They also determined soluble CEA by ELISA in the cultures of these cells. The relationship between CEA expression level and the antitumor effect of anti-CEA-CAR-T cells was evaluated using a functional assay for various PDAC cell lines. Authors observed a significant correlation between CEA expression and IFN-Ù§ secretion in vitro by ELISA; thus, the antitumor effect. Further, they created orthotopic PDAC xenograft mouse models with PDAC cell lines and injected with anti-CEA-CAR-T, only the cell lines with high CEA expression exhibited a significant therapeutic effect. Further they have identified that retrospective analysis of pathological findings from PDAC patients showed a correlation between the intensity of CEA immunostaining and tumor heterogeneity.

Because human clinical trials of CEA-targeting CAR-T cells for different solid tumors are ongoing, this study adds a significant amount of new information in this area of research. The manuscript was written in good language for better understanding of the wide spectrum readers and overall architecture of the manuscript is in good standard.

While the efforts of the authors are truly appreciated in conducting this study, I have following suggestions in their experimental approaches to improve the outcome of this manuscript. I propose some minor changes for the better manuscript output which authors may or may not consider/agree.

  1. In Fig. 1a, the authors have showed the CEA expression in different PDAC cell lines by immunofluorescence, flow cytometry and western blotting. Showing a basal expression comparison with a normal counterpart would add more information. Since CEA expression levels are key to CEA-targeting CAR-T therapy, adding information on the same in decent cohort of human clinical samples (with control matches) will give insights into the statistical significance of this study in relation to the PDAC patients.
  2. Is there any possible explanation why IFN-Ù§ secretion is more in MKN45 cells compared to Capan-1 cells in ELISA assay while the CEA expression is showing reversely in Western blotting and Immunofluorescence.

Author Response

Response to Reviewer 2 Comments

1.

In Fig. 1a, the authors have showed the CEA expression in different PDAC cell lines by immunofluorescence, flow cytometry and western blotting. Showing a basal expression comparison with a normal counterpart would add more information. Since CEA expression levels are key to CEA-targeting CAR-T therapy, adding information on the same in decent cohort of human clinical samples (with control matches) will give insights into the statistical significance of this study in relation to the PDAC patients.

Response 1:

Thank you for your suggestion. I agree that CEA expression in patient specimens should be quantified and further evaluated for reactivity with CAR-T. However, those could only be analyzed retrospectively using patient specimens, in this study. We include it for the future study.

2.

Is there any possible explanation why IFN-Ù§ secretion is more in MKN45 cells compared to Capan-1 cells in ELISA assay while the CEA expression is showing reversely in Western blotting and Immunofluorescence.

Response 2:

Thank you for this comment. As you pointed out, a discrepancy with the result of FCM was observed in some cells in the evaluation of cellular CEA by Western blotting. We speculate that this discrepancy was an overestimation of CEA because it was detected not only on the cell membrane but also in the cell endogenous CEA protein by Western blotting. Please refer to Page 7, L284-288.

Molecules on the surface of the cell membrane are the target antigens for CAR-T, so there was a correlation trend between the number of CEA molecules on cell membranes and IFN-γ secretion. Please also refer to Figure 3B.

We have now also added the sentense in the Results section to avoid any potential confusion: ' CAR-T recognize target antigen of tumor cells directly.…' (Page 7, Lines 279-281).

Reviewer 3 Report

The paper form Sato and colleagues describes the development of an anti-CEA CAR to target pancreatic cancer. They herein proposed CEA as a key target for PDAC and show that their CEACAR is sensitive to the level of CEA at the cell surface, however, they also claimed that CEACAR is sensitive to high CEA serum levels. Finally, they tested three orthotopic xenograft models and conclude that CEACAR could be used in the clinic.

The reviewer does not find the story very clear; CEA is not express in all tumours and low levels are not detected by their CAR constructs, which is also sensitive to serum levels of CEA, but still, it is described as a “successful” CAR. In addition, the article does not answer the question about safety and TME infiltration of such a CEACAR T cell product (introduce at the beginning). The target expression in pancreatic cancer is already accepted, what did they show that was not known in this topic? Their construct was already validated (PMID: 27757303) but the data presented here are not really supporting efficacy. Finally, the reviewer has concerns about the reproducibility of the data, statistics tests and number of replicates are not clearly indicated. For these reasons the reviewer recommend rejection.

Major:

-         Figure 1 the authors use three different methods (confocal, WB and flow) to detect the expression of CEA in cell lines, it might have brough more information to test different antibodies on only one assay, for example flowcytometry which clearly depict what the CAR will see. Why not use the F11-33 antibody to perform a test.

-         Figure 3: Confirmation of the activation of the CAR co-cultured with PDAC cell lines using ELISA (INFg) – no surprise but the number of donors is unclear and the authors should show the level of expression

-         Figure 4. Cytotoxicity and INFg in the presence of soluble CEA – Soluble CEA does not have a huge impact the CAR efficacy – higher cons. shows some reduction (text says “suppression of cytotoxicity”). Number of biological replicates (different donors?) unclear.

-         Figure 5 – Reduction in the 3xPC3 cell line compared to the other cells lines brings the authors to conclude that their construct is efficient (or “successful” as in the title), but two other cell lines are not sensitive. All mice were killed at day 21, the reduction small, are four mice sufficient to conclude that the CAR T cells control the tumour? The title is completely misleading and not supported by the results.

-         In the discussion, the authors state that they verify that the CAR was safe – but this was not demonstrated? They should have tested it on some healthy tissue. They also state that they see that there are limitations to the safety assay. One can see a reduction of the tumour compared to the ctrl but does the CAR control the tumour or would it have relapsed if they let it grow longer than 21d. 

Minor:

-         NGMC should be mock T cell, and the control cell should be treated as the CAR cells, which means virus transduced

-         L. 74: what is “CD3z in the external domain”?

-         L. 257: why use only S.E.M.?

-         Discussion: a tandem CAR against CEA and mesothelin has been shown to overcome the off-target toxicity (PMID: 30103775): any comments?

Author Response

Response to Reviewer 3 Comments

Major:

-         Figure 1 the authors use three different methods (confocal, WB and flow) to detect the expression of CEA in cell lines, it might have brough more information to test different antibodies on only one assay, for example flowcytometry which clearly depict what the CAR will see. Why not use the F11-33 antibody to perform a test.

   Response 1:
We appreciate for this supportive comment.
I assume you are referring to F11-39, which I would like to include in future study.
With respect to our results, F11-39 is a clone against moiety for membrane anchoring, and the CEA antibody used in this study (COL-1) is also a clone against membrane-bound CEA. It is known that COL-1 does not react with nonspecific cross-reacting antigen (NCA) and with human polymorphonuclear leucocyte, and therefore we consider that there was no significant difference in target recognition of anti-CEA-CAR in this experiment.
Thank you again for these precious suggestions.

-      Figure 3: Confirmation of the activation of the CAR co-cultured with PDAC cell lines using ELISA (INFg) – no surprise but the number of donors is unclear and the authors should show the level of expression

Response 2:
We thank you for this comment. The donor used in this study was one healthy donor. Please also refer to Page 4, Line 182.
The expression rate of the CARs used was stable at approximately 20% each time. In the treatment experiments on mice, we used CAR-T purified and tested with an expression rate of over 90%. Please also refer to Supplementary Figure S2.

Figure 4. Cytotoxicity and INFg in the presence of soluble CEA – Soluble CEA does not have a huge impact the CAR efficacy – higher cons. shows some reduction (text says “suppression of cytotoxicity”). Number of biological replicates (different donors?) unclear.

   Response 3:
Thank you for this comment. The donor used in this study was one healthy donor. Please also refer to Page Page 4, Line 182.
When high concentrations of soluble CEA (1000 ng/ml) were added, the cytotoxicity of CAR-T against BxPC-3 and MKN45 was significantly suppressed, but the serum CEA of pancreatic cancer patients is 3.51 ± 2.34 ng/mL and therefore we consider there is hardly effect in clinical practice by serum CEA. Please also refer to Page 17, Lines 516-521.

Figure 5 – Reduction in the 3xPC3 cell line compared to the other cells lines brings the authors to conclude that their construct is efficient (or “successful” as in the title), but two other cell lines are not sensitive. All mice were killed at day 21, the reduction small, are four mice sufficient to conclude that the CAR T cells control the tumour? The title is completely misleading and not supported by the results.

Response 4:
Thank you so much for this insightful comments.
We fully agree that “Successful tumor shrinkage~” was a hype and misleading expression, so we accordingly changed the title to “Successful tumor growth suppression of pancreatic cancer orthotopic xenograft model by CEA-targeting CAR-T cells“.

-      In the discussion, the authors state that they verify that the CAR was safe – but this was not demonstrated? They should have tested it on some healthy tissue. They also state that they see that there are limitations to the safety assay. One can see a reduction of the tumour compared to the ctrl but does the CAR control the tumour or would it have relapsed if they let it grow longer than 21d. 

Response 5:
We appreciate for this supportive comment.
As for Safety, we mentioned that there were no serious side effects, at least there were no significant difference in cytokine secretion and no significant weight loss in Page 17, Lines 523-531. But we were not able to examine pathology, so it is limitation.
Since this experiment was an orthotopic transplantation model of human pancreatic cancer cells into mice, anyway the safety issue of the orthotopic mouse model itself was unavoidable because of the possibility of graft versus host disease (GVHD). In addition, a report (PMID: 29903803) stated that orthotopic PDAC mouse models experiment showed a decline in body condition requiring euthanasia starting at about day 28 after tumor transplantation surgery.

If they let it grow longer than day 21, we presume that tumor suppression probably continued to some period due to the persistence of CAR-T.

Minor:

-         NGMC should be mock T cell, and the control cell should be treated as the CAR cells, which means virus transduced     

Response 6:
Thank you for your comment. We agree on your point and will include it in future studies.

  1. 74: what is “CD3z in the external domain”? 

    Response 7:
    Thank you for this comment. We corrected our mistake.
    L75 “CD3z” to “CD3ζ”.  “external domain” to “signaling domain”

-       

  1. 257: why use only S.E.M.?    

    Response 8:
    Thank you for this comment. We used S.E.M. to estimate the population mean and to compare the population means. It is to show how accurate the mean is in repeated experiments.

    Discussion: a tandem CAR against CEA and mesothelin has been shown to overcome the off-target toxicity (PMID: 30103775): any comments?

    Response 9:
    Thank you for this suggestion. We consider that there is still room for the development of novel mechanisms in CAR-T therapy, such as Tandem CAR-T, which must be incorporated in the future. On the other hand, while off-target effects could be mitigated in Tandem CAR-T, we consider that there may be limitations depending on the amount of target antigen, as in this study. We consider that the selection of cases is an important factor as well as the development of novel CAR-T itself. Accordingly, we added the sentence and references. (Page 2, Lines 84-86. Reference 6)